# Roller Speed Skating Kinematics and Electromyographic Analysis: A Methodological Approach

**DOI:** 10.3390/sports10120209

**Published:** 2022-12-14

**Authors:** Giulia Bongiorno, Helena Biancuzzi, Francesca Dal Mas, Giuseppe Fasano, Luca Miceli

**Affiliations:** 1Physiotherapist, Friuli Riabilitazione, 33080 Roveredo in Piano, PN, Italy; 2Department of Pain Medicine, IRCCS C.R.O. National Cancer Institute of Aviano, 33081 Aviano, PN, Italy; 3Department of Management, Ca’ Foscari University of Venice, 30121 Venice, VE, Italy; 4Bio-Engineer Independent Researcher, 87100 Cosenza, CS, Italy

**Keywords:** inertial sensor, groin injury, surface analysis, tailored rehabilitation

## Abstract

Roller speed skating is a discipline similar to hockey and ice skating from a biomechanical point of view, but there are no specific functional protocols for rehabilitation and performance improvement for these athletes. The aim of the study is to create a dedicated functional, kinematic and electromyographic protocol to be used as a tool for future studies on the subject. The protocol was created, starting from a correct and repeatable movement as a case study, on a world speed skating champion, using an inertial sensor positioned at the level of the first sacral vertebra, eight electromyographic probes positioned on one or the other lower limb, and a high-definition camera at 50 Hz. The results show the electromyographic activity of the muscles investigated, the degree of absolute muscle activation and compared to their maximum voluntary isometric contraction (MVIC), the level of co-activation of the agonist/antagonist muscles, and the accelerations of the body on the three axes of space. The results will represent the basis for physiotherapy and specific training use. Future developments will include the analysis of a sample of elite athletes to be able to build a normal range on the parameters investigated, and the possibility of treating in the most appropriate way possible muscle injuries (which mostly occur in the groin in such athletes) once they have occurred, even with oriented MVIC or co-activation oriented exercises.

## 1. Introduction

Speed skating is a discipline that involves cyclical and repetitive movements that are particularly tiring for elite athletes, making them susceptible to injuries, especially at the groin level. The movements of the speed roller skater are similar to those of hockey players and ice skaters, so it is reasonable to assume a similar risk in high-level athletes. In hockey, about 10% of injuries can be traced back to groin problems [1,2,3].

Throughout a season, it was found that about 20% of professional athletes had an injury not due to trauma, with significant consequences on the competitive season due to long recovery times [4,5].

Considering that the contact surface between the athlete and the ground is reduced compared to the surface of the sole of the foot, both in the case of the skate with wheels and with an ice blade, with consequent potential lateral-medial instability, the mediolateral muscles support of the hip, knee and ankle is particularly stressed. Furthermore, the hip movements in the anterior advancement in the sagittal plane are multiplanar (on the three planes) so that the blade or the wheels are subject to accelerations on the three axes of space. For this reason, in-depth knowledge of the activity of the muscles of the lower limb can be of crucial importance in the prevention of injuries in this district and any specific rehabilitation process. Compared to running, the propulsion of skating requires a much greater lateral excursion of the thigh, leg, and foot [6]. In particular, at the hip, propulsion is achieved by explosive extension, abduction, and external rotation [7].

It has long been hypothesized that muscular strains, particularly in the adductor region, may be caused by repetitive eccentric contractions that attempt to decelerate the leg during a step [8,9,10].

Previous studies of muscle activity during ice hockey have indicated that the gluteus maximus and vastus muscles are active during the propulsion phase to extend, abduct, and externally rotate the hip and extend the knee, respectively [11]. It is also reported that the hamstrings are more active during the sliding phase (isometric phase) of skating to increase the stiffness of the knee joint with the coactivity of the knee extensors [12]. Additionally, the hamstring shows high activity even during the propulsion phase in ice skating when hip extension occurs [1].

The activity of the anterior tibial is maximum during the sliding phase to stabilize the ankle and during the recovery phase to flex the ankle dorsally [13]. Furthermore, it is plausible that the range of motion and acceleration of the hip increase with higher speeds, thus increasing the response to arousal, total effort and eccentric load on the adductor muscles. The very nature of the propulsion in the skating stride at high-speed skating can predispose the hockey player and/or wheel and ice skater to strain injuries and/or prolong recovery after injury [14].

To our knowledge, no functional protocols have been specifically developed for the study of non-traumatic accidents in the field of roller skating, which, despite having clear similarities with ice skating and hockey, could present peculiar aspects not yet investigated.

The aim of the study is to elaborate on the basis of the analysis of an elite female athlete, world champion in wheel speed skating, an electromyographic and kinematic analysis protocol of the lower limb during straight-line skating, with the aim to provide athletic trainers and physiotherapists with an analysis tool to prevent non-traumatic accidents and build personalized rehabilitation programs in this branch. This protocol is able to analyze the phases of the skating cycle in terms of duration, the level of activation of the muscles of the lower limb, their degree of co-activation during the two phases, propulsion and recovery, of the skating, acceleration and speed reached by the body during these phases.

## 2. Materials and Methods

### 2.1. Participants

The participant in the study is an elite athlete (woman, 30 years old, 50 kg, 160 cm), a former Italian, European, and world champion in speed roller skating. Subject in a state of good health, no muscle-tendon, joint or other clinical pathologies.

### 2.2. Measures

Surface EMG was used to measure muscle activity during the skating test (EMG model “freemg 1000”, sampling frequency 1000 Hz). The electrodes (diameter 24 mm, Kendall Arbo^®^) were applied to predetermined muscles of the right leg (respecting the guidelines of the Seniam project) [15]. The muscles analyzed were soleus, gluteus maximus, gluteus medius, adductor magnus, rectus femoris, biceps femoris, vastus lateralis and tibialis anterior. In the region of application of the electrodes, the skin was shaved, lightly rubbed with sandpaper and cleaned with alcohol. The electrodes were fixed with adhesive tape. At this point, the participant performed maximal isometric contractions to collect EMG data for amplitude-normalization. To normalize the EMG, the following steps were performed: (1) push against a wall while standing on the sole of the foot (2) dorsal flexion of the foot in a sitting position with manual resistance (3) extension of the seated knee with a fixed shank (4) flexion of the knee in a prone position with manual resistance (5) hip flexion against manual resistance in supine position (6) hip extension in prone position (7) hip abduction against manual resistance lying on side (8) hip adduction in semi-sitting position against a roller of foam held between the thighs (the exercises were performed as presented by Kartineen et al. [14]). The participant performed maximum voluntary isometric contraction (MVIC) for EMG amplitude-normalization for three seconds. The EMG activities of each muscle recorded during skating were normalized to the maximum values obtained during these contractions regardless of the task in which the maximum values were obtained.

EMG signals coming from 1000 Hz elecromyograph were full-wave rectified, and amplitudes were scaled to maximum voluntary contractions (MVIC with 3 s contractions were performed). Time series were low-pass filtered using a zero-lag 4th order Butterworth algorithm at a cut-off frequency of 10 Hz. 

Accelerometer: a triaxial accelerometer (200 Hz, G sensor, BTS Bioengineering, Corp., Garbagnate Milanese, Italy) was positioned at the S1 level of the subject, medially, fixed by means of the supplied adjustable strap.

Camera: a high-definition camera at 50 frames/s (Vixta 50, BTS bioengineering, Corp., Garbagnate Milanese, Italy) was positioned on the sagittal plane in order to frame the subject both during the outward course on the straight track and on the way back. The camera was synchronized with both the inertial sensor (accelerometer) and the electromyographic probes. The analysis software used was EMG analyzer (BTS bioengineering, Corp., Garbagnate Milanese, Italy). The skating speed of the subject was measured using a TrueCam optical device (50 Hz, Eltraff, srl, via Torquato Tasso, 46, 20863 Concorezzo, MB, Italy) positioned next to the HD camera.

### 2.3. Design and Procedures

The study was performed at an outdoor skating rink, on a straight line of about 100 m, with a total length of the oval of about 300 m. The participant performed a warm-up before taking the measurements. At this point, the eight electromyographic probes were connected (BTS bioengineering, Corp., Garbagnate Milanese, Italy), first on the right lower limb for the first acquisitions and then on the left lower limb for the following acquisitions. The connection with the PC was wireless. In order not to disturb the athlete during the movement with the eight probes installed at the same time, it was decided to divide the acquisitions into four times, using four probes at a time (group 1 right, group 2 right, group 1 left, group 2 left) and then repeating this four times (round trip along a straight line which corresponds to about 10 skating cycles), with a rest of about 5 min between one acquisition and another. The muscles have been divided as follows: group 1 vastus lateral muscle, long head of the hamstring, tibialis anterior and soleus; group 2 gluteus maximus, rectus femoris, gluteus medius and adductor longus. For each task, MVIC was performed with the four probes in place as described above and then, without moving them, video, electromyographic and kinematic recording was performed. The duration of each motor task (round trip) was approximately 90 s. The data was then collected and analyzed in order to generate a semi-automatic reporting system.

### 2.4. Statistical Analysis

A total of four acquisitions were performed along the straight of the skating rink, two for each lower limb, integrating and synchronizing the electromyographic and kinematic data described above, for which the values indicated in the report are the average values of the aforementioned cycles. The data coming from the electromyograph, those coming from the accelerometer and those coming from the camera are synchronized from a temporal point of view within the same software (EMG-analyzer) in order to obtain electromyographic information and frame by frame accelerations, and the standard deviations are reported added to the mean values.

## 3. Results

The data acquisition of the professional athlete made it possible to build a functional protocol proposal containing a series of kinematic and electromyographic information relating to skating speed on wheels in a straight line. The high-definition video footage of the camera was used to divide the skating cycle into the two phases of propulsion (phase 1) and recovery (phase 2), capturing the exact moment of contact between the wheels and the ground and their detachment for differentiating them, integrating the EMG data and those coming from the inertial sensor. The muscles most closely correlated with the video analysis of contact and detachment of the skate (the final protocol, in order to be used in the future as extensively and semi-automatically as possible, will not include the use of the HD camera) were the vastus lateralis in the group 1 (whose activation and deactivation coincided with the aforementioned phases) and the gluteus medius muscle in group 2, for which these were chosen as the start and stop signals of the cycle (which always started with the support of the right and ended with the next support of the right skate) and as an index of subdivision of the cycle between the propulsion and recovery phases (the deactivation of these muscles, on the one hand, marked the end of the propulsion phase on that side and the beginning of the propulsion phase on the opposite side, i.e., the right propulsion coincided with the left side recovery and vice versa). It was thus possible to describe the times of the two phases, on a minimum of five cycles analyzed for each acquisition. The following Figure 1 and Figure 2 show the athlete during the trial. 

The Trucam was able to provide the test execution speed in real-time (photo) or as an average of multiple instantaneous measurements.

The EMG data were used to describe the contribution of the eight muscles investigated (in two groups of four on each side as described above) during the skating (Figure 3 and Figure 4 were reported as an example). The protocol developed for each of them is able to indicate the absolute electrical activity (in Figure 3) or indexed activity on the respective MVICs in the different phases of the cycle, with the percentage contribution of each muscle investigated to the movement, as well as the trends of the accelerations on the three axes, the mean values of the paired coactivations of the agonist/antagonist muscles investigated according to Rudolph [16] and the dynamic trend of coactivations according to Ranavolo [17] (Rudolph was used to obtain a numerical value of coactivation between muscle pairs while Ranavolo allowed us to observe coactivation dynamically during the skating cycle). The reporting system is also able to indicate the time duration of the two phases of the cycle (propulsion and recovery), as well as the percentage contribution of the muscles investigated to the movement, total and in the two phases of the cycle.

We can see how our protocol is also able to describe the phases of the cycle, which at the athlete’s average speed (24 km/h in test conditions) bring the complete cycle at a duration of about 2 s, with a slightly longer duration of the propulsion phase compared to the recovery phase (51% versus 49%). The accelerations on the vertical axis in the coronal plane are less marked than those on the other two axes, with the maximum values being reached on the anterior-posterior one (with maximum values approaching 15 m/s^2^). It is also possible to evaluate the muscular contribution during the two phases, both in absolute terms and in% of MVIC (in Figure 4, Figure 5 and Figure 6 by way of example one of the four reports generated by the system). An average and dynamic analysis of the coactivations is also generated, in which the specific example appears more marked during the propulsion phase in the muscles examined.

## 4. Discussion, Limitations, and Conclusions

To our knowledge, our protocol is the first complete functional protocol for speed wheel skating. The data provided by the protocol may, in the future, be used to objectively measure and offer rehabilitators electromyographic data such as the activation of certain muscles and the co-activation of antagonist muscles; in fact, we know that a lower degree of co-activation, for example, between gluteus maximus and the rectus femoris during the recovery phase leads to greater speed in ice skating and this, if confirmed on the road, could guide post-injury physiotherapy in elite athletes [14]. In the same work, emphasis is also placed on the importance of having a low activation of the gluteus maximus in the recovery phase, all important information for the muscular physiotherapist to rehabilitate the athlete after an injury. The adductor magnus, on the other hand, tends to be activated to a greater extent depending on the speed of ice skating, and this too may be helpful, if confirmed, in the prevention of groin injuries, which are frequent in athletes of this discipline [1]. Therefore, an athlete who has electromyographic values on a certain muscle or group of muscles constantly elevated with respect to his MVIC may be invited to strengthen the muscle in question more; in the case of coactivations that deviate too much from the reference values, they can lead to a post-injury training/rehabilitation program also aimed at optimizing this ratio, which can be measured over time at pre-established time intervals. The protocol may also be useful to explore the field of muscle asymmetries with their potential predisposition to injury, comparing the right lower limb and left lower limb, and remembering that in speed skating, both on the rink and on ice, the athlete always turns in the right direction, counterclockwise, with an ever-present muscle asymmetry [18,19,20]. However, there is little possibility of obtaining information relating to muscle fatigue with this protocol setting and having limited information on intra- and inter-muscular coordination with the exception of data relating to coactivations.

From the acquisitions in Figure 3, we can see as an example how gluteus maximus and gluteus medius are activated more in the propulsion phase, in a manner consistent with other works performed on ice, and this is in favor of applicability of the protocol also in this discipline as well as on wheels [14]. 

In the case of field hockey, recent works indicate that rehabilitation exercises with higher percentage muscle activation are preferable compared to MVIC of the respective muscles in elite female athletes, and the protocol is able to offer useful information also for this scope [21].

The protocol also has some limitations: it was built on a single subject, although world champion and therefore with movements and muscle activations considered optimal, so it will be necessary to create terms (regulatory parameters) of reference in a more extensive way with other professional athletes of both genders, in the same discipline, also to investigate any gender and anthropometric differences on performances. Acceleration also refers to the body “as a whole” and not only to the lower limb since the inertial sensor was positioned medially at the level of S1. Although it is reasonable to assume that the protocol may be beneficial not only for speed roller skating but also for hockey and ice skating, it will also be tested on professional athletes belonging to the latter two disciplines. The protocol will also be tested at increasing standardized speeds (e.g., on a dedicated treadmill) to assess the impact of speed on the analysis. During the design of the functional protocol, we introduced the need to acquire MVIC only for the amplitude-normalization of the sEMG signal with the aim of better estimating the signal amplitude during the skate task in relation to a reference. We did not use the MVIC for the study of motor co-ordination and for the estimation of muscle fatigue. The myoelectric manifestation of muscle fatigue, estimated by measuring the decrease in fiber conduction velocity, which is reflected in an amplitude increase and spectral compression over time, can be analyzed predominantly in the frequency domain. In order to estimate the muscle fatigue estimation in isometric conditions, by avoiding doing it during movement where the signal is no longer considered stationary, we should provide two fatigue sessions in isometric conditions before and after the skating task but, at this stage, we are interested in a protocol that is easy to use in the field. 

We began to include muscle co-activation as the first indicator of motor co-ordination. Further analyzing muscular behavior in relation to kinematic behavior would be optimal. At this early stage, we have begun to propose a simple functional protocol without the need to make the athlete wear a whole-body sensor network. At the moment, the protocol still offers the possibility of analyzing muscular behavior within defined events through the use of kinematic signals acquired with the IMU. Video recording also always allows for a visual inspection of what is happening on the track. 

Finally, better measurements should definitely use a camera of at least 100 Hz.

In conclusion, our protocol was developed with several purposes: the first is to offer data to the athletic trainers of street roller skating an easily obtainable kinematic and electromyographic database (a set of acquisitions and data processing requires less than one hour of time and can be transportable anywhere, since all analysis components are equipped with rechargeable batteries) to build training programs based on precise outcomes (acceleration, sEMG).The second is to offer physiotherapists a tool capable of following the rehabilitation path of the injured athlete over time, working not only on the kinematic outcomes but also on the electrical signals of the muscular activation (degree of absolute muscle activation and compared to MVIC, degree of co-activation of the muscles agonist-antagonists). 

The major strength of our protocol in the opinion of the authors, is to be able, using the EMG-analyzer software, to generate the complete report for each group of four muscles in a few minutes in a semi-automatic way: for the intervention in fact, the operator is only required to define, on an EMG basis, the beginning and end of the analysis period, to confirm the beginning of the phases of the EMG-based cycles—beginning of the electrical activity of the gluteus medius and vastus lateral muscles that automatically identifies—and to identify, again on an sEMG basis, the transition from the propulsion phase to the recovery phase when the electrical activity of the aforementioned muscles is switched off, with the possibility of adjustments thanks to video analysis and accelerations synchronized with sEMG. These phases have been deliberately left in manual mode to make the protocol as malleable as possible, leaving the possibility of modulating it, for example, on ice athletes, in which the transition from the propulsion phase to the recovery phase could be slightly different. This is obtainable by comparing the video analysis with the electromyographic data, even better in the future if associated with kinematic behaviors (angles between body segment).

Once the system has been calibrated on ice mode, video analysis will no longer be necessary, in a similar way to asphalt, and the same protocol with the two calibrations can be used for ice and asphalt, providing physiotherapists and kinesiologists with an additional tool for studying similarities and differences of two similar but not equal disciplines in their kinematics and patterns of muscle activation.

## Figures and Tables

**Figure 1 sports-10-00209-f001:**
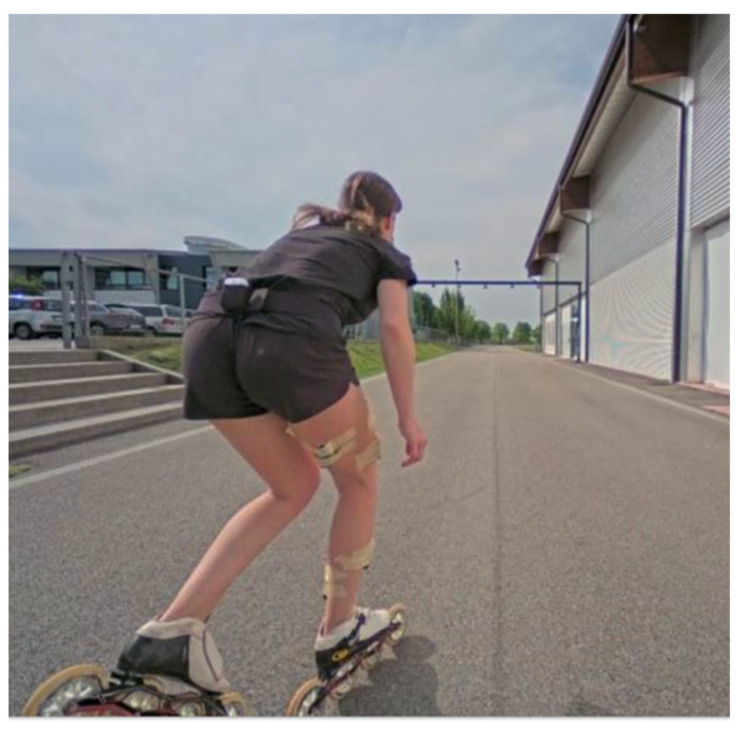
Sample frame used to calibrate G sensor to split out propulsion and recovery phases during skating. Permission obtained by participator.

**Figure 2 sports-10-00209-f002:**
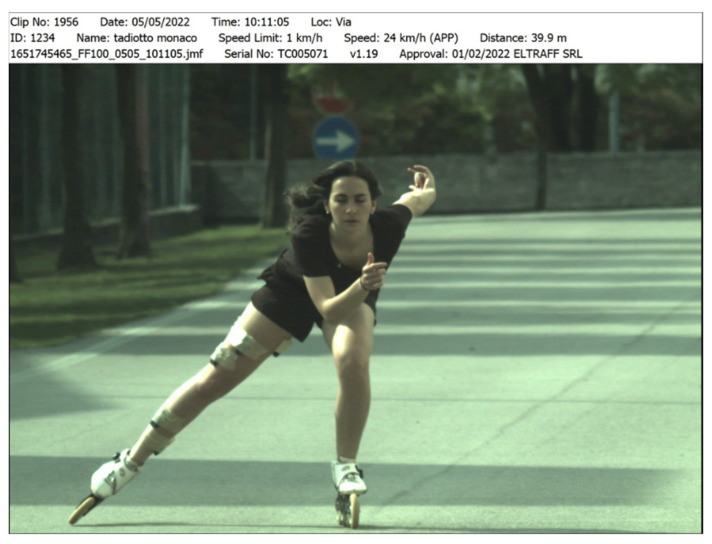
Frame used to detect speed skate ride with Trucam (first line on the bottom of the figure, expressed in km/h). Permission obtained by participator.

**Figure 3 sports-10-00209-f003:**
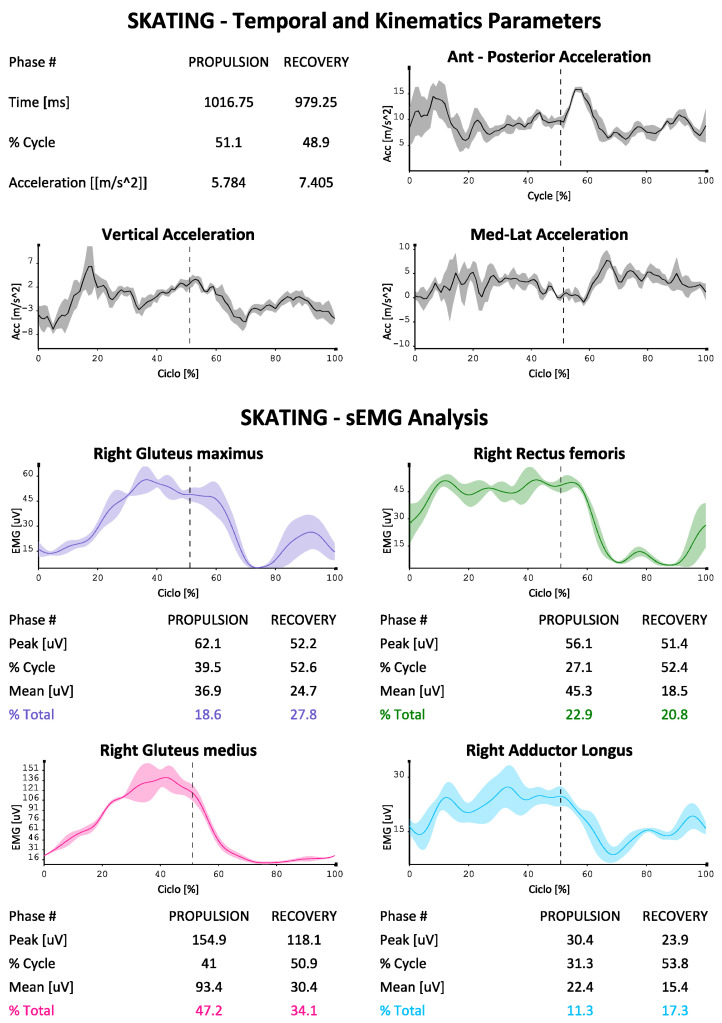
EMG signals in first group of four muscles sample during propulsion and recovery phases (cycle percentage time in X axis, mV developed in y axis), and percentage contribution to the movement for each muscle, temporal percentages of the phases of the cycle.

**Figure 4 sports-10-00209-f004:**
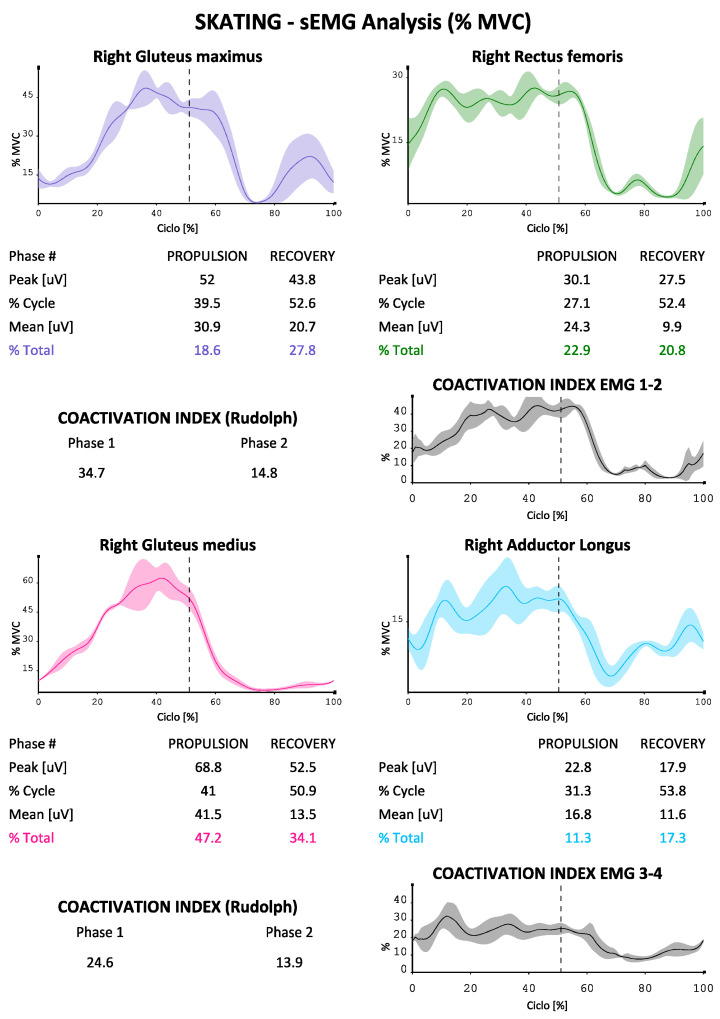
EMG signals in first group of four muscles sample during propulsion and recovery phases (cycle percentage time in X axis, mV developed in y axis), and percentage contribution to the movement for each muscle, temporal percentages of the phases of the cycle.

**Figure 5 sports-10-00209-f005:**
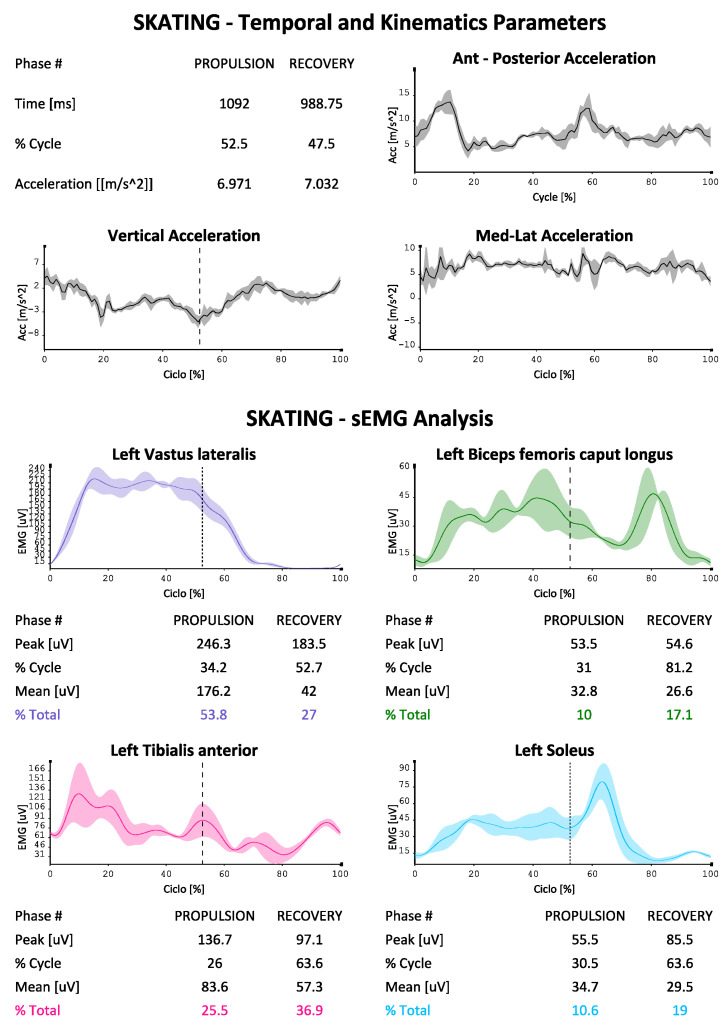
EMG signals in second group of four muscles sample during propulsion and recovery phases (cycle percentage time in X axis, mV developed in y axis), and percentage contribution to the movement for each muscle, temporal percentages of the phases of the cycle.

**Figure 6 sports-10-00209-f006:**
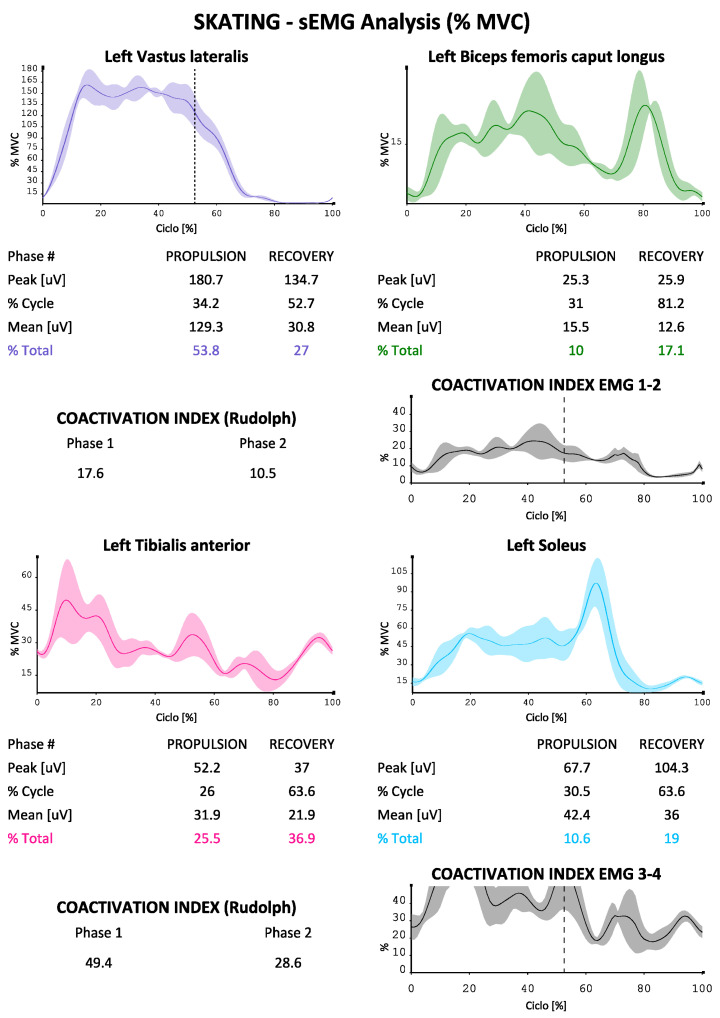
EMG signals in second group of four muscles sample during propulsion and recovery phases (cycle percentage time in X axis, mV developed in y axis), and percentage contribution to the movement for each muscle, temporal percentages of the phases of the cycle.

## Data Availability

The full dataset can be obtained from the corresponding author upon reasonable request.

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
