# Peer review of "Roller Speed Skating Kinematics and Electromyographic Analysis: A Methodological Approach"

_sports, 2022, doi:10.3390/sports10120209_

Round 1

Reviewer 1 Report

The authors of this article have defined a functional protocol for 'roller speed skating' based on kinematic, surface electromyographic and video information.

The purpose of this protocol is to allow for proper testing of training and rehabilitation interventions.

From a general point of view, the protocol is well designed and the text is well written and easy to understand.

Any kind of statistical inference is absent. That is why I would suggest including in the title that it is, for example, a 'case repor' or a 'methodological approach'. For the same reason I would avoid including a 'last name' in the title in order not to create false expectations before reading.

Specific comments

Line 66: “To our knowledge, no clinical protocols have been developed specifically for the study of non-traumatic accidents in the field of roller skating …”. I would replace clinical with functional.

Line 77: “… recovery, of the skating …”. I would delete the comma.

Line 92: “…to collect the EMG normalization data.” I would use the following sentence: to collect EMG data for amplitude-normalization. Throughout the text I would replace 'normalization' with 'amplitude-normalization'.

Line 93: It would be useful to indicate the muscle of interest for each type of exercise performed for MVIC.

Line 106: “(1 second average of 3-second contraction).” I did not understand it very well, but if I did, it would be wrong. I would ask you to better explain how the reference value for normalization is calculated.

Line 115: “EMG – Analyzer”: delete the hyphen between the two words.

Line 116: walking? I thought you were investigating skating.

Line 127: I would avoid using the term 'phases' because they get confused with skating phases.

Lines 129: “… about 100 meters …”. Already written, delete it.

Lines 139-142: I would rewrite this section. Information on synchronisation between devices should not be reported in this section. It should also be specified that standard deviations are reported in addition to the mean.

Line 160: sn?

Lines 175-177: Please explain better. It is not clear what Rudolph was used for and what Ranavolo was used for.

Line 190: “The so called …” with “the so called …”.    

Author Response

We are grateful for the opportunity of revising our work.

Here is a point-to-point response to the reviewer’ requests. All the changes can be seen in the revised version of the manuscript underlined in yellow.

We do hope that the new version of the manuscript will meet everyone’s expectations.

Best regards,

The Authors

Reviewer 1

Any kind of statistical inference is absent. That is why I would suggest including in the title that it is, for example, a 'case repor' or a 'methodological approach'. For the same reason I would avoid including a 'last name' in the title in order not to create false expectations before reading.

Thanks for your suggestion. We agree. As recommended, we amended the title and avoided the last name throughout the paper.

 Line 66: “To our knowledge, no clinical protocols have been developed specifically for the study of non-traumatic accidents in the field of roller skating …”. I would replace clinical with functional.

Done. Thank you.

Line 77: “… recovery, of the skating …”. I would delete the comma.

Done. Thank you.

Line 92: “…to collect the EMG normalization data.” I would use the following sentence: to collect EMG data for amplitude-normalization. Throughout the text I would replace 'normalization' with 'amplitude-normalization'.

Done. Thank you.

Line 93: It would be useful to indicate the muscle of interest for each type of exercise performed for MVIC.

The exercises were performed as presented by the Kartineen et al. paper. (Kaartinen S, Venojärvi M, Lesch KJ, Tikkanen H, Vartiainen P, Stenroth L. Lower limb muscle activation patterns in ice-hockey skating and associations with skating speed. Sport Biomech. 2021;1–16.)

Line 106: “(1 second average of 3-second contraction).” I did not understand it very well, but if I did, it would be wrong. I would ask you to better explain how the reference value for normalization is calculated.

better explained, thank you.

Line 115: “EMG – Analyzer”: delete the hyphen between the two words.

Done. Thank you.

Line 116: walking? I thought you were investigating skating.

Modified. Thank you

Line 127: I would avoid using the term 'phases' because they get confused with skating phases.

We now use “times”. Thank you.

Lines 129: “… about 100 meters …”. Already written, delete it.

Done. Thank you.

Lines 139-142: I would rewrite this section. Information on synchronisation between devices should not be reported in this section. It should also be specified that standard deviations are reported in addition to the mean.

Integrated. Thank you.

Line 160: sn?

Modified. Thank you.

Lines 175-177: Please explain better. It is not clear what Rudolph was used for and what Ranavolo was used for.

Integrated. Thank you.

Line 190: “The so called …” with “the so called …”.    

Done. Thank you.

Reviewer 2 Report

Dear Authors,

 It is a nice pilot-study you have performed, in a quite challenging environment.

I’m missing some data in the paper: muscles of the lower leg, Vastus lateralis and Hamstrings. The Vastus lateralis is several times mentioned as trigger muscle for the detecting the two phases.

It would be great if you could add these graphs: e.g. as supplement and may think if you would optimize the graphs instead of just using a screenshot. Because there is so much more information in the graphs if you place them according to information you give in the text.

Further, is missing Type of EMG-System and its sampling rate, sampling rate of accelerometer and TrueCam.

Comment to MVIC, body position, Frame rate of camera:
MVIC is a possibility to define the muscular activation, but it can’t give any information about the inter and intra-muscular coordination during a movement and also limited information about possible fatigue effects during a movement, e.g. in the frequency domain.

 Further, the use of the EMG system without kinematic Information (angles between body segment). I’m not sure if this is possible, as the movement pattern may change, e.g. upper body position is different like more bend or more upright or the hip and knee angles are different to the reference subject.

50Hz is low frame rate for the camera, in speed-ice skating it is too low, because the movement of the clap-skate is too fast. Therefore, I'm wondering if 100Hz or even more would be better in further studies.It would be nice to add this in your paper.

Comments line by line

Line 39: lateral -lateral?  Should this be lateral-medial?

Line 53/54: Question: What is about the co-activation of the internal and external rotation muscles in order to block any rotation in the foot?

Line 83: Type of EMG and Sampling rate

Line 96 to 99: twice the same information in two sentences.

Line 106: Sampling rate

Line 114: walking speed better skating speed

Line 192: “on the vertical axis are less marked than those”, choose better wording

Line 226-228: see also comment about MVIC above.

Line 248/249: “the kinematic outcomes but also on the electrical ones (degree of absolute muscle activation and compared to MVIC, degree of co-activation of the muscles agonist-antagonists)

the electrical ones: choose better wording, e.g. electrical signals of the muscular activation

Line 263-267: see also comment about body position.

Author Response

We are grateful for the opportunity of revising our work.

Here is a point-to-point response to the reviewers’ requests. All the changes can be seen in the revised version of the manuscript underlined in yellow.

We do hope that the new version of the manuscript will meet everyone’s expectations.

Best regards,

The Authors

Reviewer 2

I’m missing some data in the paper: muscles of the lower leg, Vastus lateralis and Hamstrings. The Vastus lateralis is several times mentioned as trigger muscle for the detecting the two phases.

It would be great if you could add these graphs: e.g. as supplement and may think if you would optimize the graphs instead of just using a screenshot. Because there is so much more information in the graphs if you place them according to information you give in the text.

Thanks for your interest! we added some images.

Further, is missing Type of EMG-System and its sampling rate, sampling rate of accelerometer and TrueCam.

Done, Thank you.

Comment to MVIC, body position, Frame rate of camera:

MVIC is a possibility to define the muscular activation, but it can’t give any information about the inter and intra-muscular coordination during a movement and also limited information about possible fatigue effects during a movement, e.g. in the frequency domain.

During the design of the functional protocol we introduced the need to acquire MVIC only for the amplitude-normalisation of the sEMG signal with the aim of better estimating the signal amplitude during the skate task in relation to a reference. We did not use the MVIC for the study of motor co-ordination and for the estimation of muscle fatigue. As commented by the reviewer, the myoelectric manifestation of muscle fatigue, estimated by measuring the decrease in fiber conduction velocity, which is reflected in an amplitude increase and spectral compression over time, can be analysed predominantly in the frequency domain. In order to estimate the muscle fatigue estimation in isometric conditions, by avoiding doing it during movement where the signal is no longer considered stationary, we should provide two fatigue sessions in isometric conditions before and after the skating task but, at this stage, we are interested in a protocol that is easy to use in the field. We have added this aspect within the limits of the study.

In contrast, we began to include muscle co-activation as the first indicator of motor co-ordination.  

 Further, the use of the EMG system without kinematic Information (angles between body segment). I’m not sure if this is possible, as the movement pattern may change, e.g. upper body position is different like more bend or more upright or the hip and knee angles are different to the reference subject.

Here again, we absolutely agree with the reviewer. Analysing muscular behaviour in relation to kinematic behaviour would be optimal.

In contrast, we have begun to propose a simple functional protocol without the need to make the athlete wear a whole-body sensor network. At the moment, the protocol still offers the possibility of analysing muscular behaviour within defined events through the use of kinematic signals acquired with the IMU. Video recording also always allows a visual inspection of what is happening on the track. However, we have included these considerations within the 'limitations of the study.

50Hz is low frame rate for the camera, in speed-ice skating it is too low, because the movement of the clap-skate is too fast. Therefore, I'm wondering if 100Hz or even more would be better in further studies.It would be nice to add this in your paper.

The authors have implemented the limitations. Thank you.

Line 39: lateral -lateral?  Should this be lateral-medial?

Modified, thank you

Line 53/54: Question: What is about the co-activation of the internal and external rotation muscles in order to block any rotation in the foot?

The authors did not analyze this aspect. it is a useful starting point for future studies. Thank you

Line 83: Type of EMG and Sampling rate

Added. Thank you.

Line 96 to 99: twice the same information in two sentences.

Modified. Thank you.

Line 106: Sampling rate

Modified. Thank you.

Line 114: walking speed better skating speed

Modified. Thank you.

Line 192: “on the vertical axis are less marked than those”, choose better wording

Modified. Thank you.

Line 226-228: see also comment about MVIC above.

Integrated. Thank you.

Line 248/249: “the kinematic outcomes but also on the electrical ones (degree of absolute muscle activation and compared to MVIC, degree of co-activation of the muscles agonist-antagonists)

Done. Thank you.

the electrical ones: choose better wording, e.g. electrical signals of the muscular activation

Done. Thank you.

Line 263-267: see also comment about body position.

Integrated. Thank you.
